Cloning and evaluation of reference genes for quantitative real-time PCR analysis in Amorphophallus

Wang Kai
Niu Yi
Wang Qijun
Liu Haili
Jin Yi
Zhang Shenglin konjac@163.com
1 College of Horticulture and Landscape, Southwest University , Chongqing , China
2 Ministry of Education, Key Laboratory of Horticulture Science for Southern Mountainous Regions, Southwest University , Chongqing , China
3 Chongqing Education Commission, Konjac Resource Utilization Engineering Research in Chongqing Colleges and Universities , Chongqing , China
Verger Alexis
Electronic publication date: 2017 Apr 26
Publication date: 2017
Volume: 5
Electronic Location ID: e3260
Received 2016 Oct 6; Accepted 2017 Apr 1
Copyright: ©2017 Wang et al.
Copyright year: 2017
Copyright holder: Wang et al.
License: This is an open access article distributed under the terms of the Creative Commons Attribution License, which permits unrestricted use, distribution, reproduction and adaptation in any medium and for any purpose provided that it is properly attributed. For attribution, the original author(s), title, publication source (PeerJ) and either DOI or URL of the article must be cited.
License URL: https://creativecommons.org/licenses/by/4.0/

Keywords: Gene expression, Real-time reverse transcription PCR, Reference genes, Amorphophallus, Monocotyledon

Funding: National Natural Science Foundation of China 31071796 “111” Project B12006 “121” konjac scientific and technological support demonstration project of Chongqing 104190/40605014 This project was supported by grants from the National Natural Science Foundation of China (31071796), the “111” Project (B12006), and the “121” konjac scientific and technological support demonstration project of Chongqing (104190/40605014). The funders had no role in study design, data collection and analysis, decision to publish, or preparation of the manuscript.

==============================
Quantitative real-time reverse transcription PCR (RT-qPCR) has been widely used in the detection and quantification of gene expression levels because of its high accuracy, sensitivity, and reproducibility as well as its large dynamic range. However, the reliability and accuracy of RT-qPCR depends on accurate transcript normalization using stably expressed reference genes. Amorphophallus is a perennial plant with a high content of konjac glucomannan (KGM) in its corm. This crop has been used as a food source and as a traditional medicine for thousands of years. Without adequate knowledge of gene expression profiles, there has been no report of validated reference genes in Amorphophallus. In this study, nine genes that are usually used as reference genes in other crops were selected as candidate reference genes. These putative sequences of these genes Amorphophallus were cloned by the use of degenerate primers. The expression stability of each gene was assessed in different tissues and under two abiotic stresses (heat and waterlogging) in A. albus and A. konjac. Three distinct algorithms were used to evaluate the expression stability of the candidate reference genes. The results demonstrated that EF1-a, EIF4A, H3 and UBQ were the best reference genes under heat stress in Amorphophallus. Furthermore, EF1-a, EIF4A, TUB, and RP were the best reference genes in waterlogged conditions. By comparing different tissues from all samples, we determined that EF1-α, EIF4A, and CYP were stable in these sets. In addition, the suitability of these reference genes was confirmed by validating the expression of a gene encoding the small heat shock protein SHSP, which is related to heat stress in Amorphophallus. In sum, EF1-α and EIF4A were the two best reference genes for normalizing mRNA levels in different tissues and under various stress treatments, and we suggest using one of these genes in combination with 1 or 2 reference genes associated with different biological processes to normalize gene expression. Our results will provide researchers with appropriate reference genes for further gene expression quantification using RT-qPCR in Amorphophallus.

Introduction

Gene expression analysis is an effective and widely used approach to elucidate transcriptional regulatory networks and identify novel genes (Thompson, Regev & Roy, 2015). In comparison with other techniques that are used to measure transcript abundance, quantitative real-time reverse transcription-PCR (RT-qPCR) has become the preferred method for gene expression studies because of its speed, sensitivity, and specificity (Bustin, 2000; Bustin et al., 2009; Nolan, Hands & Bustin, 2006). There are many factors that affect the accuracy of RT-qPCR data, including amplification efficiency (Pfaffl, 2005) and RNA quality (Vermeulen et al., 2011). Selecting a suitable reference gene and using a set of standardized experimental conditions are among the most important factors for quantifying gene expression by RT-qPCR (Bustin et al., 2009; Hellemans & Vandesompele, 2014). Suitable reference genes have been identified for many crops and especially for model plants (Dekkers et al., 2012; Czechowski et al., 2005; Itoh et al., 2016). Reference genes that are used as appropriate normalization genes usually have high copy numbers and relatively stable expression (Thellin et al., 1999). These genes are ubiquitously expressed and are typically involved in housekeeping processes (Pfaffl et al., 2004).

Ideal reference genes should be expressed stably in different environments, different tissues, different growth stages, and different experimental conditions, and their expression should not be significantly different between samples (Li et al., 2005). However, an increasing number of studies have found that commonly used reference genes do not typically satisfy all these criteria: their expression typically shows large variations between different types of cells and tissues, different stages of organ development and cell proliferation, different culture conditions in vitro, and different experimental conditions (Lee et al., 2002; Hisbergues et al., 2014; Dheda et al., 2004; Zmienko et al., 2015). Selection of suitable reference genes largely depends on seeking specific reference genes with stable expression in each examined cell or tissue and each experimental condition (Hellemans & Vandesompele, 2014).

In recent years, studies have selected reference genes for different experimental conditions in an increasing number of crops, such as watermelon (Kong et al., 2014), cotton (Wang, Wang & Zhang, 2013), tomato (Løvdal & Lillo, 2009), switchgrass (Huang et al., 2014), rice (Maksup, Supaibulwatana & Selvaraj, 2013), wheat (Paolacci et al., 2009), pearl millet (Saha & Blumwald, 2014), and Oenanthe javanica (Jiang et al., 2014). Many reference genes have been identified as suitable reference genes under specific experimental conditions. For instance, ACT7 was the best reference gene in pearl millet and O. javanica (Jiang et al., 2014) under abiotic stress, EF1-a was the most suitable reference gene in chrysanthemum aphid infestation (Gu et al., 2011) and TUA, UBI-ep, and EF1-a data could be used to normalize RT-qPCR data in cucumber (Wan et al., 2010).

Amorphophallus, a perennial herb in the family Araceae, genus Amorphophallus, is mainly found in India, the Indo-China Peninsula, south China (mostly in Yunnan province) and Southeast Asia (Gille et al., 2011). Its underground corm has abundant konjac glucomannan (KGM), and this species is the only crop that is able to produce abundant KGM in nature (Fang & Wu, 2004). KGM is a high-molecular-weight, water-soluble dietary fiber (Zhang, Xie & Gan, 2005) with a variety of clinical and health effects on the human body. Furthermore, some phenolics and alkaloids in Amorphophallus have a beneficial effect in the treatment of chronic diseases. Because of these properties, Amorphophallus is a rare and special vegetable and medicinal plant that has been widely grown in southwest China, and KGM is used in food, medicine, and other fields. In recent years, with the increase of the emphasis on health problems, the use of konjac flour as a health food and medicine has been more attentively studied. Overall, Amorphophallus, the only plant that can produce large amounts of KGM, is an important resource crop in Asia and throughout world. Interestingly, Amorphophallus is a typical shade-demanding plant, intolerant to heat damage and waterlogging, and flooding and high temperature are two important factors that significantly impact Amorphophallus growth. Understanding the expression level of genes associated with resistance and elucidating the expression patterns of some key regulatory genes would contribute to genetic improvement of Amorphophallus. However, as one of the main identification methods of gene determination, RT-qPCR requires stable reference genes. Therefore, verification of stable reference genes in Amorphophallus is necessary. So far, due to limitations in genomic data, there have been no studies that screened for suitable reference genes in Amorphophallus. In this study, we selected 9 candidate reference genes that are commonly used in other crops: glyceraldehyde-3-phosphate dehydrogenase (GAPDH), elongation factor 1- α (EF1-a), the eukaryotic initiation factor 4A (EIF4A), H3.2 histones (H3), cyclophilin (CYP), β-actin (ACTB), β-tubulin (TUB), ubiquitin (UBQ), and ribosomal protein L16 (RP), (Yang et al., 2014; Huang et al., 2014; Gopaulchan, Lennon & Umaharan, 2013). In addition, the expression levels of the SHSP gene in leaves, which is upregulated in response to heat stress in plants, were verified to determine the suitability of the candidate reference genes. The purpose of this study is to determine whether the candidates are suitable reference genes under different conditions by comparing the expression stability of these genes in Amorphophallus, which would provide a theoretical basis for the identification of the expression of target genes in related regulatory networks.

Materials and Methods

Plant materials and treatments

Two types of cultivated species (Amorphophallus albus and Amorphophallus konjac) with a high konjac glucomannan (KGM) content and good quality from the Xiema resource nursery (Xiema, Beibei, Chongqing, PR China) were used as experimental materials. The corms were planted in pots containing soil and peat (2:1, V/V), cultured in a greenhouse and irrigated once every 7 days. After the blades were fully extended, the plants were treated with a simulated high temperature (40 °C) or waterlogging (2.5 cm above the soil surface), and the leaves were harvested after 0 h, 2 h, 4 h, 8 h, and 1 d of exposure (Gu et al., 2011). Different tissue samples (leaves, roots, and corms) were harvested from the same plant after being excavated from the pot and rinsed with distilled water. The materials for verifying the expression profile of SHSP were treated at 42 °C and harvested at 0 h, 1 h, 2 h, 4 h, and 8 h. All samples were immediately frozen in liquid nitrogen after harvest and stored at −80 °C prior to use.

RNA isolation and first-strand cDNA synthesis

Total RNA was extracted from Amorphophallus leaves using the ZH120 Quick RNA Islation Kit (Waryong, Beijing, China). RNA was treated with RNase-free DNase I (Waryong, Beijing, China). Total RNA from roots and ground corms was extracted by the pine method (Chang, Puryear & Cairney, 1993; Gille et al., 2011). The concentration and contaminants were measured using a NanoDropTM 2000 spectrophotometer (Thermo Scientific, USA), and only samples with an OD260/280 between 1.8 and 2.0 and OD260/230 >2.0 were used for cDNA synthesis. The quality and integrity of total RNA were verified by 1% agarose gel electrophoresis. First-strand cDNA synthesis was conducted by using a PrimeScript ™ RT reagent Kit RR047A with gDNA Eraser (TaKaRa, Japan) and generated by using random hexamers. One microgram of total RNA from each sample was used in reverse transcription reactions according to the manufacturer’s instructions. The synthesized cDNAs were diluted to 25-fold for RT-qRCR analyses.

Table 1 8 candidate reference genes’ accession numbers of 4 monocots in GenBank.

Name	Anthurium andraeanum	Elaeis guineensis	Phoenix dactylifera	Musa acuminata	
EF1-a	JN602204.1	XM_010939210.2	XM_008792592.1	XM_009404789.2	
TUB	JN602205.1	XM_010907949.2	XM_008789128.2	XM_009404284.2	
ACTB	JN602202.1	XM_010932692.2	XM_008783179.2	XM_009392480.2	
H3	No find	XM_019852580.1	XM_008813893.2	XM_009414503.2	
CYP	JN602201.1	XM_010942984.2	XM_008801198.2	XM_009414618.2	
GAPDH	JN602203.1	XM_010912103.1	XM_008802722.2	XM_009389227.2	
EIF4A	JN602200.1	XM_010927143.2	XM_008799912.2	XM_009417246.2	
UBQ	JN602199.1	XM_019854937.1	XM_008811850.2	XM_009393249.2	

Gene cloning

Degenerate primers of 8 candidate reference genes were designed based on conserved regions of homologous sequences of different monocots, such as Anthurium andraeanum, Phoenix dactylifera, Elaeis guineensis, and Musa acuminata. The accession numbers from GenBank of 4 monocots are shown in Table 1. The accession numbers of the primer sequences are listed in Table 2. The amplified PCR products were A-T cloned into the pEASY-T1 simple vector, and then, the products were transformed into T1 competent cells. The pEASY-T1 simple vector and T1 competent cells were used with a pEASY-T1 simple cloning kit (Transgen, Beijing, China). Positive colonies were identified using colony PCR. The M13 forward and reverse primers were used to recover polymorphisms of the amplification results. Bacterial liquid cultures of positive clone products were sequenced by the Beijing Genomics Institute (BGI, China), and the sequences were identified as the candidate homologous reference genes using NCBI BLAST (Table 3). All of the sequences of the 8 candidate reference genes as well as the published RP sequences (254998327) are shown in Sequence S1.

RT-qPCR assay

RT-qPCR reactions were executed in 96-well plates using the Bio-Rad CFX96 Real-Time PCR system (Bio-Rad, USA) and SsoFast EvaGreen Supermix (Bio-Rad, USA). The total volume comprised 4 ng of cDNA template, 0.2 μM reverse primer, 0.2 μM forward primer, 5 μL of SYBR Green mix, and ddH2O to 10 µL. RT-qPCR reactions were conducted using the following parameters: 95 °C for 30 s followed by 40 cycles of 95 °C for 5 s and 55−65 °C for 5 s. At the end of the process, the specificity of the amplification was tested using melting curve analysis from 65 to 95 °C with a 0.5 °C increase in temperature at each step. Each RT-qPCR reaction set included water as a negative no-template control for each primer pair. To obtain more reliable experimental results, each PCR assay was conducted in triplicate (technical replicates).

Table 2 Degenerate primers used to clone 8 candidate reference genes for RT-qPCR in Amorphophallus.

Candidate reference genes	Forward primer sequence [5′–3′]	Reverse primer sequence [5′–3′]	
EF1-a	GACTGCCACACCTSMCAYA	CKACRCTCTTGATRACACCR	
TUB	CGCCCYGACAACTTCGTCTT	CTTGGMRTCCCACATCTGCTG	
ACTB	AYGACATGGAGAARATYTGGC	CACCAGARTCMARCACAATAC	
H3	SGTSAAGAAGCCCCACC	RCGRGCAAGCTGGATGT	
CYP	GAGAACTTCMGGGCNCTC	AYCTGSCCGAACACGACG	
GAPDH	CAADGACAAGGCTGCDGCTCA	CTTKGCDGCACCAGTGCTGC	
EIF4A	GATGAGCTNACCCTTGAGGGT	GCTGDACATCAATACCACGAGC	
UBQ	TRACGGGBAAGACCATCACN	ACCTTGTARAACTGGAGGASDGCG	

Table 3 Description of Amorphophallus candidate reference genes for RT-qPCR.

‘Source’ represents which species these genes of different accession number belong to.

Name	Description	Accession number	Source	Length (bp)	Identity (%)	
EF1-a	Elongation factor 1-α	XM_015774249.1	Oryza sativa	241	87	
TUB	β-tubulin	L33263.1	Oryza sativa	636	85	
ACTB	β-actin	XM_015784227	Oryza sativa	231	83	
H3	Histone H3.2	XM_015784228.1	Oryza sativa	283	90	
CYP	Cyclophilin	XM_008680450.1	Zea mays	317	86	
GAPDH	Glyceraldehyde-3-phosphate dehydrogenase	XM_015780140.1	Oryza sativa	334	86	
EIF4A	Eukaryotic initiation factor 4A	NM_001111926.1	Zea mays	304	87	
UBQ	Ubiquitin	NM_001138130.1	Zea mays	301	94	

Data analysis

The methods commonly used to analyze the stability of reference genes are geNorm (Vandesompele et al., 2002), BestKeeper (Pfaffl et al., 2004), and NormFinder (Andersen, Jensen & Orntoft, 2004), which are based on different computer algorithms to rank candidate reference genes under different experimental conditions (De Spiegelaere et al., 2015). The geNorm, BestKeeper, and NormFinder programs that were used to calculate the stable values were downloaded for experimental analysis (e.g., geNorm version 3.5 (http://medgen.ugent.be/ jvdesomp/genorm/), BestKeeper version 1 (http://www.gene-quantification.de/bestkeeper.html), NormFinder version 0.953 (http://www.mdl.dk/publicationsnormfinder.htm)).

The expression levels of the candidate reference genes were measured according to their quantification cycle (Cq) values. For geNorm and NormFinder, the raw Cq values were converted to the required data using the formula: 2−△Cq (△Cq = each corresponding Cq value-minimum Cq value; Liu et al., 2012). The geNorm algorithm was used to further determine the expression stability of the candidate reference genes by calculating the average expression stability value (M) and pairwise variation (V) between all pairs of genes. The ranking of candidate reference genes is based on those with lower M values. In addition, the BestKeeper algorithm used the untransformed Cq values to analyze stability.

Normalization of SHSP gene

The gene expression levels of SHSP were quantified at different time points under heat treatment using the best four reference genes (EF1-a, EIF4A, H3, and, UBQ) separately and a combination of multiple reference genes (EF1-a+H3, EF1-a+UBQ, and EF1-a+H3 +UBQ). Two traditional reference genes (ACTB and GADPH) were also selected for normalization to control for different normalization factors. The software qBasePlus version 3.0 (http://www.biogazelle.com/) was downloaded for these calculations (Hellemans et al., 2007).

Results

Verification of the primer specificity and PCR amplification efficiency

Specific primers for the reference genes and SHSP were designed using Primer3Plus (Untergasser et al., 2007; http://primer3plus.com/) based on the results of sequencing and the published ribosomal protein (RP) sequences of Amorphophallus. The product sizes of the 9 candidate reference genes and SHSP amplified by specific primers were between 100 and 200 bp in length. Each primer pair was evaluated by the presence of a single peak in melting curve analysis (Fig. S1A) and a single, distinct band on a 3% agarose gel (Fig. S1B). The gene-specific PCR amplification efficiency (E) and correlation coefficient (R2) were calculated using a standard curve in which another replicate was performed using a standard curve generated by 10-fold serial dilutions of gel-extracted PCR products. The amplification efficiency (E) was calculated as follows: E = (10−1∕slope − 1) × 100% (Pfaffl, 2001). Only primers with an ideal value range (110% ≥ E ≥ 90%) and correlation coefficient (R2 ≥ 0.99) were used for subsequent experiments. The primer sequences and amplification characteristics, including the Tm, length, efficiency, and R2, are shown in Table 4. The PCR amplification efficiencies (E) of nine candidate reference genes and SHSP were between 92.8% and 109.5%. Standard curve regression equation correlation coefficients (R2) were between 0.990 and 0.999, which demonstrated a strong linear relationship.

Table 4 Description of 9 candidate reference genes and SHSP gene in Amorphophallus.

Tm represents melt temperature and was calculated by the Bio-Rad CFX96 Real-Time PCR system.

Name	Forward primer sequence [5′–3′]	Reverse primer sequence [5′–3′]	Amplicon length (bp)	Tm (°C)	E (%)	R2	
RP	GGACGAAGAGCAATGACCC	ACCCTTTCCCCGAACCCA	118	79.5	92.8	0.991	
EF1-a	AAGTTCCTGAAGAATGGCGAT	GTCCCTCACGGCAAACCTACC	111	82.5	99.8	0.990	
TUB	GCTGGTTGAGAATGCCGATGAA	GCAGAAATAAGGTGATTGAGAT	120	80	97.7	0.998	
ACTB	CCAACAGAGAGAAGATGACA	ACCAGAATCCAGCACAATAC	128	79	94.3	0.999	
H3	CGGGAGATCGCTCAGGACT	CATGATGGTGACGCGCTTG	139	86.5	96.8	0.990	
CYP	CAAGCCCCTCCACTACAAGG	CCGGTGTGCTTCTTCACGAA	153	86	93.0	0.991	
GAPDH	ACTAACTGCCTCGCTCCTC	CAGCCCTTCCACCCCTCCA	145	82	93.3	0.990	
EIF4A	ACAAGATGAGGAGCAGGG	GGTGATAAGGACACGAGA	116	79.5	109.5	0.990	
UBQ	GGACACCATCGACAACGTGA	TTCTTCTTGCGCTTCTTGGC	189	87.5	107.7	0.997	
SHSP	ATCAAGGTCCAGGTGGAGGA	GGCAGCGAGAACTTCCTCAT	131	88	96.1	0.996	

Cq values of candidate reference genes

The raw Cq values of 9 candidate reference genes ranged from 18.73 (RP) to 34.78 (ACTB; Table S1), and the mean Cq value was used for further analysis. The mean Cq values of the reference genes were between 22.78 (RP) and 27.96 (TUB). The threshold fluorescence for TUB was slightly higher than that of the other genes, indicating that TUB had a low level of expression. In general, there were large variations in the expression levels of each of the 9 reference genes (Fig. 1).

geNorm analysis

geNorm is a Microsoft-based VBA macro software that was developed by Vandesompele et al. (2002). The basic principle is that the ratio of the expression levels of two ideal reference genes under any experimental conditions or cells should be identical in all samples, keeping the calculated M value of a single gene as low as possible. In addition, an increasing number of studies have found that using two or more reference genes contributes to correct system deviation and obtaining more reliable results (Bustin et al., 2009; Vandesompele et al., 2002). Although geNorm can use standardized factors to pair differences, obtaining a lower threshold is preferable.

Figure 1 Threshold cycle values (Cq) of 9 candidate genes across 78 cDNA samples in RT-qPCR.

The lower and upper ends of each box represent the 1/4 and 3/4 quartiles. Whiskers represent the maximum and minimum Cq values. The median Cq values are depicted by the dots in the boxes.

We ranked the 9 candidate reference genes in 7 sets according to their expression stability values from low to high (Table 5). The threshold of the M value was 1.5; genes with an M value under 1.5 could be considered to be reference genes. EF1-a and EIF4A were two of the best three reference genes in all sets, with values under the threshold in each case. In A. konjac heat-treated samples, the values of all of the candidate reference genes were under the threshold value, and H3 had the lowest value (M = 0.78). UBQ performed well under heat stress in all heat-treated samples. Under waterlogged conditions, TUB was the highest-ranked gene in the two species besides EF1-a and EIF4A, and 9 and 5 candidate reference genes could be regarded as reference genes based on their M values in A. albus and A. konjac, respectively. In a comparison of the expression of the 9 candidate reference genes across different tissues, only EF1-a and EIF4A were under the threshold value in the two species of Amorphophallus. Although UBQ was the best candidate reference gene for A. albus, it was unstable in A. konjac. The M value of CYP was above 1.5 in A. konjac, but it ranked highly in the two species across different tissues. Therefore, we could regard it as an available additional reference gene across different tissues. “Total” contained all test samples, and EIF4A and EF1-a were the best two reference genes in the total set. The values of UBQ and H3 were also under 1.5. The two most stable genes calculated by geNorm at each step during stepwise exclusion of the least stable reference gene are shown in Fig. 2. Starting from the least stable gene at the left, the genes are ranked according to increasing expression stability, ending with the two most stable genes on the right. In general, EIF4A and EF1-a showed remarkable stability in all sets. At the same time, H3 and UBQ under heat stress, TUB under waterlogging, and CYP across different tissues were also expressed stably. In order to avoid using different genes belonging to the same biological process as reference genes, we excluded EIF4A and repeated the geNorm analysis. The results showed that EF1-a was still one of the stable ones in all sets (Fig. 3).

Table 5 Expression stability values for 9 candidate genes calculated using geNorm.

Aa represents A. albus, and Ak represents A. konjac. Total set contains all test samples.

Rank	Heat in Aa	Heat in Ak	Waterlogging in Aa	Waterlogging in Ak	Tissues in Aa	Tissues in Ak	Total	
1	EF1-a (1.07)	H3 (0.78)	EF1-a (0.66)	EF1-a (1.27)	UBQ (1.33)	EIF4A (1.43)	EIF4A (1.24)	
2	EIF4A (1.12)	EIF4A (0.82)	EIF4A (0.71)	EIF4A (1.31)	EF1-a (1.49)	EF1-a (1.50)	EF1-a (1.28)	
3	UBQ (1.16)	EF1-a (0.83)	TUB (0.72)	TUB (1.44)	EIF4A (1.50)	CYP (1.53)	UBQ (1.42)	
4	GAPDH (1.19)	UBQ (0.89)	RP (0.74)	UBQ (1.49)	CYP (1.50)	RP (1.79)	H3 (1.42)	
5	ACTB (1.26)	ACTB (0.90)	H3 (0.74)	RP (1.50)	H3 (1.64)	ACTB (1.79)	ACTB (1.52)	
6	H3 (1.28)	TUB (1.07)	ACTB (0.98)	ACTB (1.57)	RP (1.64)	UBQ (1.99)	CYP (1.59)	
7	CYP (1.35)	CYP (1.10)	CYP (1.05)	H3 (1.62)	TUB (2.01)	TUB (2.01)	RP (1.59)	
8	TUB (1.84)	GAPDH (1.17)	UBQ (1.14)	GADPH (1.74)	ACTB (2.11)	H3 (2.11)	TUB (1.60)	
9	RP (1.90)	RP (1.32)	GAPDH (1.45)	CYP (1.75)	GADPH (3.25)	GAPDH (2.35)	GAPDH (1.93)	

Figure 2 Gene expression stability (M) and ranking of potential reference genes within different treatment groups as calculated by geNorm.

The ordinate value represents the average expression stability value (M) and the abscissas show the 9 candidate reference genes. A lower M value represents more stable expression as analyzed by geNorm algorithm in different sets, including heat stress in A. albus (A), heat stress in A. konjac (B), waterlogging in A. albus (C), waterlogging in A. konjac (D), different tissues in A. albus (E), different tissues in A. konjac (F), and total (G). The total set contains all test samples.

Figure 3 Gene expression stability (M) and ranking of potential reference genes within different treatment groups as calculated by geNorm when EIF4A was excluded.

The ordinate value represents the average expression stability value (M) and the abscissas show the 9 candidate reference genes. A lower M value represents more stable expression as analyzed by geNorm algorithm in different sets, including heat stress in A. albus (A), heat stress in A. konjac (B), waterlogging in A. albus (C), waterlogging in A. konjac (D), different tissues in A. albus (E), different tissues in A. konjac (F), and total (G). The total set contains all test samples.

Furthermore, to determine the optimal number of reference genes required for effective normalization, pairwise variation (Vn/Vn+1), which is required for normalization between sequential normalization factors (NFs), was introduced by geNorm. All of the results of pairwise variation are illustrated in Fig. 4. The cut-off value set by the algorithm is 0.15, below which the inclusion of an additional reference gene is not required. For example, V2/3 of A. konjac in heat stress and V4/5 in A. albus under waterlogging were under the cut-off values, which indicated that 2 and 4 reference genes were required for more reliable normalization in these two conditions, respectively. While the 0.15 threshold was not met in the other samples, the geNorm developers emphasized in the user manual, that the proposed threshold of 0.15 must not be taken as a strict cutoff. The cut-off value was set only to offer guidance for determining the optimal number of reference genes. Therefore, they recommend that using only the 3 best reference genes is, in most cases, a valid normalization strategy and results in a much more accurate and reliable normalization compared to the use of only a single reference gene.

Figure 4 Determination of the optimal number of reference genes for normalization by pairwise variation by geNorm.

Aa represents A. albus, and Ak represents A. konjac. Total set contains all test samples.

NormFinder analysis

Claus et al. compiled the NormFinder program in 2004, and the principle of the program is similar to that of geNorm. NormFinder generates a stable value for gene expression, then sorts gene expression in ascending order according to the stable value (Andersen, Jensen & Orntoft, 2004). The gene with the lowest stable value is the most stable reference gene. However, this program has the drawback that it identifies only a single best reference gene.

The results calculated by NormFinder were similar to those calculated by geNorm. All of the values, ranked from low to high, are shown in Table 6. The lower the value, the higher the stability. In A. albus under heat stress, EF1-a was the best reference gene, followed by EIF4A. In heat-treated A. konjac, H3 was the best reference gene, followed also by EIF4A. Under waterlogging conditions, the results showed that EF1-a and EIF4A were the best two reference genes in the two species. Additionally, UBQ and RP performed well across different tissues in A. albus based on the results calculated by NormFinder. In the total set, EIF4A and EF1-a showed remarkable expression stability in the two species of Amorphophallus. Overall, EIF4A and EF1-a performed very well in all sets and were identified as the best two reference genes in 5 sets.

Table 6 Expression stability values for 9 candidate genes calculated using NormFinder.

Aa represents A. albus, and Ak represents A. konjac. Total set contains all test samples.

Rank	Heat in Aa	Heat in Ak	Waterlogging in Aa	Waterlogging in Ak	Tissues in Aa	Tissues in Ak	Total	
1	EF1-a (0.32)	H3 (0.20)	EIF4A (0.09)	EF1-a (0.43)	UBQ (0.28)	EIF4A (0.41)	EIF4A (0.39)	
2	EIF4A (0.37)	EIF4A (0.27)	EF1-a (0.18)	EIF4A (0.49)	RP (0.57)	EF1-a (0.55)	EF1-a (0.47)	
3	UBQ (0.44)	EF1-a (0.28)	TUB (0.18)	TUB (0.65)	EIF4A (0.60)	CYP (0.56)	UBQ (0.63)	
4	GAPDH (0.46)	ACTB (0.37)	H3 (0.25)	UBQ (0.72)	EF1-a (0.62)	RP (0.84)	H3 (0.63)	
5	ACTB (0.55)	UBQ (0.38)	RP (0.25)	RP (0.73)	H3 (0.71)	ACTB (0.87)	ACTB (0.75)	
6	H3 (0.59)	TUB (0.57)	ACTB (0.53)	ACTB (0.80)	CYP (0.74)	UBQ (1.04)	TUB (0.83)	
7	CYP (0.67)	CYP (0.59)	CYP (0.57)	H3 (0.87)	TUB (1.03)	TUB (1.14)	RP (0.83)	
8	RP (1.12)	GAPDH (0.69)	UBQ (0.66)	GAPDH (0.96)	ACTB (1.21	H3 (1.19)	CYP (0.83)	
9	TUB (1.16)	RP (0.81)	GAPDH (0.93)	CYP (0.98)	GAPDH (2.14)	GADPH (1.41)	GAPDH (1.15)	

BestKeeper analysis

BestKeeper software was written by Pfaffl et al. (2004). The data were entered into the BestKeeper Excel file. Then, BestKeeper calculated the standard deviation (SD) and coefficient of variance (CV) of each gene. The candidate reference gene with the lowest coefficient of variance and standard deviation (CV ± SD) was considered to be the best reference gene. Any reference gene with a SD1 was excluded because gene expression was not consistent in all samples. The advantage of BestKeeper software is that it is not only able to analyze the stability of reference genes, but it can also compare the expression levels of target genes.

The results of BestKeeper analysis were different from those of the other two programs which many researches had drew similar conclusions (Lin et al., 2014; Qi et al., 2016). This may be because the principle of this algorithm differs from that of the others. Lower CV values represent higher stability. EIF4A was the best reference gene under heat stress in the two species of Amorphophallus, and ACTB and EF1-α ranked second in A. albus and A. konjac, respectively. In the waterlogging sets, even though EIF4A performed well in A. albus, it was identified as the worst reference gene in A. konjac. At the same time, EF1-α was one of the least stable reference genes under waterlogging in both species of Amorphophallus. Across different tissues, CYP, EF1-α, and EIF4A were ranked the same and were the best three genes in both species. In total, H3, TUB, and UBQ were ranked above EF1-α and EIF4A, even though EF1-α and EIF4A were ranked as unstable in other sets and were even the worst two genes in the waterlogging treatment of A. konjac. The stability of these two genes could be better than others when SD ≤ 1 was taken into consideration. In fact, this algorithm does not consider internal differences between different plants. The calculated results are only references for selecting reference genes. The ranking is shown in Table 7.

Table 7 Expression stability values for 9 candidate genes calculated using BestKeeper.

Aa represents A. albus, and Ak represents A. konjac. Total set contains all test samples.

Rank	Heat in Aa	Heat in Ak	Waterlogging in Aa	Waterlogging in Ak	Tissues in Aa	Tissues in Ak	Total	
1	EIF4A	EIF4A	EIF4A	TUB	CYP	CYP	H3	
(1.49 ± 0.46)	(1.66 ± 0.40)	(1.12 ± 0.38)	(1.29 ± 1.09)	(1.17 ± 0.96)	(1.26 ± 0.36)	(1.65 ± 1.50)	
2	ACTB	EF1-a	RP	RP	EF1-α	EF1-α	TUB	
(1.58 ± 0.84)	(1.83 ± 0.34)	(1.21 ± 0.37)	(1.80 ± 2.96)	(1.39 ± 0.67)	(1.31 ± 0.89)	(1.79 ± 0.61)	
3	EF1-α	H3	ACTB	ACTB	EIF4A	EIF4A	UBQ	
(1.62 ± 0.47)	(1.83 ± 0.30)	(1.25 ± 0.84)	(1.84 ± 2.55)	(1.59 ± 0.58)	(1.40 ± 1.08)	(1.81 ± 0.31)	
4	H3	ACTB	H3	CYP	RP	H3	EIF4A	
(1.69 ± 0.78)	(1.89 ± 0.49)	(1.47 ± 037)	(1.91 ± 1.88)	(1.62 ± 1.76)	(1.65 ± 1.24)	(1.93 ± 0.63)	
5	UBQ	GADPH	TUB	UBQ	UBQ	RP	EF1-α	
(1.94 ± 0.60)	(1.97 ± 0.99)	(1.86 ± 0.34)	(1.95 ± 0.75)	(1.92 ± 0.98)	(2.15 ± 1.27)	(1.97 ± 0.61)	
6	RP	CYP	EF1-a	H3	ACTB	GAPDH	CYP	
(1.98 ± 1.36)	(2.11 ± 0.83)	(1.97 ± 0.34)	(2.11 ± 1.35)	(2.07 ± 1.55)	(2.19 ± 1.88)	(2.06 ± 0.75)	
7	GAPDH	UBQ	UBQ	GADPH	H3	TUB	RP	
(2.04 ± 0.61)	(2.11 ± 0.49)	(3.12 ± 0.95)	(2.24 ± 1.77)	(2.13 ± 0.91)	(2.89 ± 1.20)	(2.08 ± 0.76)	
8	CYP	RP	CYP	EF1-α	TUB	UBQ	GADPH	
(2.94 ± 0.67)	(2.17 ± 1.14)	(3.52 ± 0.59)	(2.63 ± 0.67)	(2.47 ± 1.45)	(2.90 ± 1.39)	(2.32 ± 0.33)	
9	TUB	TUB	GADPH	EIF4A	GAPDH	ACTB	ACTB	
(4.03 ± 1.20)	(2.61 ± 0.61)	(6.47 ± 1.04)	(2.64 ± 0.75)	(6.90 ± 2.45)	(3.39 ± 1.22)	(2.51 ± 0.51)	

Figure 5 Relative quantification of SHSP gene in leaves at different time points after heat treatment in A. albus and A. konjac.

(A) shows the relative expression of SHSP in A. albus and (B) represents the expression level of SHSP in A. konjac. We selected the validated best reference gene(s) under heat stress and two traditional reference genes (ACTB and GADPH) as normalization factors. Asterisks (*) indicate significant differences (P < 0.05) and P value was calculated by student’s T test. Sum of squares of deviations were calculated based on the average expression level normalized by 7 identified normalization factors.

Evaluation by normalizing of SHSP

Small heat shock proteins (SHSP) are important proteins in plants that function as molecular chaperones (Jakob et al., 1993). SHSP genes in plants are involved in resistance to abiotic stress and are rapidly induced under heat stress (Sun, Van Montagu & Verbruggen, 2002). The sequences of a SHSP gene from two species were cloned and are shown in Sequence S1. The relative expression levels of SHSP analyzed using 9 different normalization factors are shown in Fig. 5. The results revealed that the trends of SHSP expression were similar by using different normalization factors. However, differences were observed at 1 h, 2 h, and 8 h and the relative expression using the identified reference genes showed almost identical results. The relative expression patterns of normalized by EIF4A and EF1-a were more similar than UBQ and H3. When ACTB and GADPH were used as reference genes, the expression of SHSP at these time points were significantly different from the results using identified stable genes in two species based on the results of student’s T test. Lower sum of squares of deviations (SS) value indicated the data was closer to the expected. The result showed SS values of 3 normalization combinations were lower than these of single identified stable reference genes.

Discussion

Amorphophallus, a poorly studied crop, is increasingly being studied because it can produce a large amount of the soluble dietary fiber konjac glucomannan (Fang & Wu, 2004). Dietary fiber is a complex mixture and is subdivided into soluble and non-soluble fiber (Chawla & Patil, 2010). Because it has good colloidal properties and excellent physiological functions, soluble dietary fiber has been widely used in food, medicine, industry, and other fields (Schneeman, 1986; Prosky et al., 1987; Farvid et al., 2016). KGM is a type of soluble dietary fiber with a high molecular weight and strong hydrophilic properties, and it has high therapeutic and medical value (Zhang, Xie & Gan, 2005). When used in food and ingested in the human digestive tract, KGM contributes to healthier digestion and the excretion of solid waste (Arvill & Bodin, 1995; Kaats, Bagchi & Preuss, 2015). Studies have also shown that KGM and some alkaloids in Amorphophallus have inhibitory effects on many diseases (Doi et al., 1979; Fan et al., 2008; Wu & Chen, 2011). Current transcriptome data for Amorphophallus have been deposited in the NCBI database, and these data accelerate the progress of studies on the important genes and in the molecular breeding of this species (Gille et al., 2011; Zheng et al., 2013). Important genes related to resistance (such as heat stress, waterlogging and other stresses) and agronomic properties could be identified by exploiting the transcriptome data and through other methods.

Gene expression analysis plays a significant role in identifying the expression profile and function of these genes (Huggett et al., 2005; Itoh et al., 2016). RT-qPCR is a preferred method to study gene expression because of its unique advantages (Bustin et al., 2009). However, without validating stable reference genes, the results of RT-qPCR data lack credibility (Hellemans & Vandesompele, 2014). In other words, the stability of reference gene expression is an elementary prerequisite for normalizing the expression profiles of target genes. Although stable reference genes are established in many crops, there have been no studies on reference genes in Amorphophallus.

The screening of reference genes in this study was carried out using different tissues and two main abiotic stresses that seriously influence the yield and quality of Amorphophallus. Waterlogging is the main factor that causes soft rot in Amorphophallus. In addition, by using different tissues, we evaluated the expression stability of the candidate reference genes across leaves, roots, and tubers under a wide range of experimental treatments.

Nine candidate reference genes whose homologs have been commonly used as reference genes in many plants were assessed simultaneously under various conditions using three distinct statistical algorithms: geNorm, NormFinder and BestKeeper. The results were mainly based on the geNorm algorithm and were further validated by NormFinder and BestKeeper. Integrating the specific identification results calculated by different algorithms, EF1-a, EIF4A, H3, and UBQ were the four most stably expressed reference genes under heat treatment. For waterlogging stress experiments, EF1-a, EIF4A, and TUB were more stably expressed than the others. In different tissues, EF1-a, EIF4A, and CYP showed the most stable expression. Overall, EF1-α and EIF4A were the best two candidate reference genes. The traditional two reference genes ACTB and GADPH had large differences in expression levels between individuals. The results illustrated the necessity of validating reference genes in Amorphophallus.

For a long time, ACTB and GADPH have been regarded as suitable reference genes, and their stable performance in many crops has been confirmed (Lin et al., 2014; Wang, Wang & Zhang, 2013). Additional studies revealed that the stability of these two genes was lower than that of other genes (Gu et al., 2011; Martins et al., 2016). However, compared to our data on gene expression and results of SHSP expression normalization, we do not recommend using them as reference genes under the conditions we tested.

Even though some candidate genes, such as H3, TUB, UBQ, and CYP, showed stable expression in a single set, they were generally less stable in other sets in our study. These genes should only be used as reference genes under certain experimental conditions or in certain organisms. In previous studies, these genes also showed stable expression in several species. For instance, H3, ACTB and UBQ have been used as internal controls for the analysis of RT-qPCR in cotton (Wang, Wang & Zhang, 2013; Huang et al., 2011). In maize, TUB was one of best reference genes for gene expression (Lin et al., 2014). UBQ2 was validated as one of most suitable reference genes across all tested samples in banana fruit (Chen et al., 2011), and CYP was one of the best scoring genes in Vitis vinifera (Borges et al., 2014). In Amorphophallus, we suggest that these genes can be used as internal controls under certain experimental conditions and may lead a reliable result when combining them with EF1-a or EIF4A.

In our study, EF1-a and EIF4A showed high stability across all treatments in two species of Amorphophallus and were identified as the best two reference genes in many sets. In fact, many studies have shown that EF1-a and EIF4A were stable in other monocots when used as reference genes. They have been selected as reference genes in perennial ryegrass (Lolium perenne L.), switchgrass (Panicum virgatum), and African oil palm (Elaeis guineensis) under different abiotic stresses (Huang et al., 2014; Gimeno et al., 2014; Xia et al., 2014). These species are typical monocots and are closely related to Amorphophallus. It has been shown that EF1-a and EIF4A are stably expressed in many monocots. For example, one recent study showed that EF1-α and EIF4A were the most stable genes in different tissues of pearl millet (Reddy et al., 2015). Additionally, EF1-a had stable expression in foxtail millet (Setaria italica L.) and maize (Kumar, Muthamilarasan & Prasad, 2013; Lin et al., 2014). It could be that the stable expression of these genes is related to the role they play in biological metabolic pathways, as both proteins are involved in protein synthesis. Elongation factors are mainly responsible for the initial steps of protein synthesis, and eukaryotic initiation factors promote polypeptide chain extension during mRNA translation (Berchtold et al., 1993; Sonenberg & Dever, 2003). This may be the main reason why EF1-a and EIF4A have similar expression in many species.

However, we were concerned that selecting multiple genes that participate in related biological processes may result in inaccurate results. geNorm algorithm works based on the assumption that the expression ratio of two ideal internal control genes is identical in all samples, regardless of the experimental conditions or cell type. This means that the absolute expression levels of these genes can change between conditions, but the ratio would be maintained. Variations of the absolute levels of each of the genes would likely reflect technical variability. This approach, however, only holds if the chosen genes belong to different functional classes; otherwise, one may be simply scoring co-regulation and may incorrectly assume that two genes are stably expressed and are appropriate reference genes, when in fact they may be responding to the treatment, but as they are part of the same process, they may be responding coordinately. When Vandesompele et al. (2002) developed this algorithm, they mentioned that special attention should be paid to selecting genes that belong to different functional classes, which significantly reduces the chance that genes might be co-regulated. To avoid the use of two reference genes that are related to the same biological process, we supposed that excluding one of them and adding 1 or 2 reference genes related to different biological processes could effectively normalize the relative expression levels of the target genes in Amorphophallus. To verify this conjecture, we excluded EIF4A and repeated geNorm analysis, and analyzed the expression of the SHSP gene by designing different normalization combinations without EIF4A in Amorphophallus under heat stress.

The results calculated by geNorm revealed that EF1-a was still one of the most stable reference genes. It probably behaves as a stably expressed gene and it is not because it is being co-expressed with another gene due to being part of the same biological process. In addition, the normalized result confirmed that the relative expression of SHSP was not significantly different when EF1-a, EIF4A, H3, or UBQ was used as a single reference gene in normalizing the gene expression profile under heat stress. Highly similar expression level of SHSP normalized by EF1-a and EIF4A suggested the presence of the co-expression of them. Although our ”best genes” are functionally related, the third and fourth-ranked ”good genes” also provided similar results compared with the two ”best genes”, which further supported the accuracy of the results calculated by three algorithms and the stability of the best four reference genes. The expression profiles of SHSP were in almost perfect agreement when normalized by three combinations (EF1-a +H3, EF1-a +UBQ, and EF1-a +H3 +UBQ). These results showed that almost no difference was found when a single identified stable reference gene was used, and using one best reference gene sometimes can obtain accurate result in RT-qPCR normalization. However, the result that 3 combinations had lower SS value probably indicate that the normalization by multiple reference genes is closer to the actual expression. Furthermore, the strategy used to select combinations of reference genes also had impact on the final results. The combination should avoid these co-regulated genes with biological processes and be evaluated by geNorm. If selected candidate genes simply vary too much to be useful in a practical manner, reliable results cannot be obtained irrespective of how many reference genes are selected.

Conclusion

In the present study, the expression of nine candidate reference genes in Amorphophallus under two major stresses and across different tissues was compared and evaluated to identify stable reference genes for gene expression studies. Among them, EF1-a and EIF4A appeared to be the two most appropriate reference genes. Integrating the analyses by three algorithms with normalization of SHSP expression, we recognized that a single reference gene may normalize the expression well under some conditions in RT-qPCR. But the result of using multiple reference genes are more credible. It is indispensable to select reference genes in a practical manner based on the specific experimental conditions and avoiding using multiple genes that participate in related biological processes. These results will provide useful information to profile the gene expression of resistance and quality-related in Amorphophallus.

Supplemental Information

Figure S1 Specificity and amplicon size of the amplification products

The primers were designed based on the conserved regions of related sequences in two species of Amorphophallus. (A) Melt curves of 9 candidate reference genes showing single peaks. (B) M represents 100 bp DNA ladder (Vazyme, Nanjing, China). A single band for each candidate gene in each species is shown in this picture. The candidate reference genes from left to right are: ACTB, EF1-α, UBQ, H3, TUB, CYP, RP, EIF4A, and GADPH. These genes in A. albus and in A. konjac are shown in odd lanes and even lanes, respectively.

Click here for additional data file.

Table S1 The Cq values of 9 candidate reference genes

Click here for additional data file.

Sequence S1 Sequences of 9 candidate reference genes and SHSP gene in two species of Amorphophallus

Click here for additional data file.

Additional Information and Declarations

Competing Interests

Author Contributions

Data Availability

The authors declare there are no completing interests.

Kai Wang conceived and designed the experiments, performed the experiments, analyzed the data, contributed reagents/materials/analysis tools, wrote the paper, prepared figures and/or tables.

Yi Niu conceived and designed the experiments, reviewed drafts of the paper.

Qijun Wang, Haili Liu and Shenglin Zhang reviewed drafts of the paper.

Yi Jin performed the experiments.

The following information was supplied regarding data availability:

The raw data is contained in Table S1.

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
