# Peer review of "Cloning and evaluation of reference genes for quantitative real-time PCR analysis in Amorphophallus"

_PeerJ, doi:10.7717/peerj.3260_

## Round 0.1 · original submission · Major Revisions

· Academic Editor

Major Revisions

Your manuscript has been assessed by three expert reviewers. Based on their reports, I am pleased to inform you that it is potentially acceptable for publication, once you have carried out very carefully some essential revisions suggested by them.

In my opinion the crucial improvement that needs to be done in this study is to analyze a few more genes.

I hope you will find all the recommendations helpful.

Reviewer 1 ·

Basic reporting

In general, the manuscript is well written and quiet clear. I have however, a few general comments on the reporting:

1) Resolution of Fig. 2 seems low.

2) Fig. S2 should be presented as a box plot, as in Fig. 1

3) In line 151, two references should be added: https://www.ncbi.nlm.nih.gov/pubmed/12184808 and https://www.ncbi.nlm.nih.gov/pubmed/19246619

4) In line 189, it should say Pfaffl et al., 2004.

5) Line 210. Maybe rephrase "promotion of good digestion" to something else, maybe "healthier digestion"?

6) Line 255. Maybe you meant "assumed" instead of "supposed"?

7) Line 255. As written, the argument makes no sense, as 18S would not be able to be used as a reference gene if oligo-dT was used as the primer for the RT reaction, as it lacks a poly-A tail.

8) Line 114. Each method should be accompanied by the proper citation.

9) Line 133. Replace E value, by Efficiency

10) The accepted abbreviation is RT-qPCR and not qRT-PCR. It is the PCR which is quantitative, not the RT (https://www.ncbi.nlm.nih.gov/pubmed/19246619). Also, please replace all instances of Ct for Cq (quantification cycle).

11) In figure legends, I would suggest to change “Aa represented Amorphophallus Albus” for “Aa stands for Amorphophallus albus”. Please mind the lower case in albus.

12) The legend in Fig. S1 appears to have been cut or something. It says “each gene set a no template control in the even slots”, which is not clear. Maybe you meant "each gene set is show in the odd lanes, plus the corresponding negative control in the even lanes"?
Also, I know what they say (I have the same machine), but the axis labels are difficult to see in Fig. S1A.

13) Fig. 1 is not cited in the text.

14) I would avoid citations in the abstract.

Experimental design

Major concerns:

1) My major concern with this study is the choice of candidate genes to test, some of which participate in related biological processes. The geNorm algorithm works based on the assumption that the expression ratio of two ideal internal control genes (i.e. two which could be used as proper reference genes) is identical in all samples, regardless of the experimental condition or cell type. This means that the absolute expression levels of these genes can change between conditions, but the ratio would be maintained. Variation of the absolute levels of each of the genes would likely reflect technical variability. This approach however, only holds if the chosen genes belong to different functional classes; otherwise, you may be simply scoring co-regulation and may incorrectly assume that two genes are stably expressed and appropriate reference genes, when in fact they may be responding to the treatment, but as they are part of the same process, they may be responding coordinately. This is mentioned in Vandesompele et al. 2002 ("Special attention was paid to selecting genes that belong to different functional classes, which significantly reduces the chance that genes might be co-regulated.").

The genes chosen in this study include 2 related to translation and 2 cytoskeleton genes, and the genes described as being the most stably expressed ones are indeed the two ones related to translation (EF1 alpha and EIF4A). In fact, in Fig. 2D, in most of the panels, the genes shown to be the most stable are these two. While it is possible that indeed, they are the most stably expressed of the ones tested, I'm concerned (and it cannot be excluded) that they were found to be the most stably expressed by geNorm simply because they are part of related processes. Although it could be argued that because two other tools, namely NormFinder and BestKeeper, gave similar (yet not identical) results (which is expected, anyway), this should minimize the concern, but the fact that geNorm was used as the main tool of the study, further being used to suggest the number of reference genes to use under different conditions, raises some concerns the authors should comment on.

I propose that new genes are included (just enough to replace the repeated functional categories), so that all genes are associated with different biological processes.

2) It appears that while the genes chosen for this study appear to work in other crops and in different systems, they, as a whole, may not be significantly stably expressed in the tested system. geNorm analysis suggested that 9 of the studied candidate genes should be used in waterlogged experiments in A. konjac! This is far from practical. This suggests that the chosen genes may exhibit significant variability under these conditions and that other genes may need to be considered. While the authors mention that they "suggest that using the top two or three ranked genes would improve the reliability of the calibration data compared with using only one single reference gene", their data, using that particular set of candidate genes, shows that is not the case, exemplified with the extreme case of the waterlogged A. konjak samples, requiring 9 genes. I would suggest that new candidate genes should be tested and checked to see if they behave better under these conditions. Check Vandesompele et al., 2002 or this recent study (https://www.ncbi.nlm.nih.gov/pubmed/27040147) for other categories to be tested.

Minor concerns:

1) Correct baseline settings are crucial for accurate Cq determinations and I’m worried of the effect of the samples with Cqs lower than 10, particularly in machines which automatically determine the baseline based on the samples (as the CFX96 used here), on the quantification of the other samples in the same run (Vandesompele et al., 2002). Samples should be diluted before analyzing the expression of the 18S gene. As long as all samples are diluted the same way, there is no problem, as these are all relative quantifications. This relates to the element raised in the Discussion (line 249). There's no need to dilute the "stock cDNAs" for this. You could prepare independent dilutions just to examine 18S levels.

2) On the other hand, very lowly expressed genes, such as CYP (which one of its Cqs was 38.63), usually show very high variability and this can affect the results (https://www.ncbi.nlm.nih.gov/pubmed/15127793). I wonder what the Cq was for the negative control in that run.

Integrating these two comments (Minor Concerns 1 and 2), please note that 18S and CYP were always the worst scoring genes. There may be a technical reason for this, rather than biological.

3) The last line of the manuscript (“more than 3 reference genes will benefit the accuracy of target gene expression profiles”), while generally a good rule of thumb for any RT-qPCR project, is not in line with the results of the paper, in which for most cases, the authors propose the use of >4 of the selected candidate genes, based on the geNorm results, for the different conditions tested.

4) I'm curious as to why the authors didn't use the transcriptome data (https://www.ncbi.nlm.nih.gov/pubmed/24759927) for designing the primers, instead of using degenerate primers based on other organisms.

5) Did the authors perform –RT controls (i.e. do qPCR on RNA samples which have not been reverse-transcribed, to assess the amount of genomic DNA contamination?). The DNAse I treatment must be verified through qPCR.

6) Which primer strategy was used for the RT step? Random primers? If so, hexamers? This should be mentioned in the methods.
In line 95, the authors mention that they used “up to 1 ug of total RNA”, which implies that they didn’t they use the same amount of RNA for all samples. Is this the case? This is a very important element to keep in mind when trying to minimize technical variation in gene expression measurements via RT-qPCR.

7) Was the cDNA diluted after the RT step and before being used in the PCR reaction? Undiluted cDNA may result in inhibition of PCR and thus, affect the quantification.

Validity of the findings

I've expressed my concerns in the previous section, which may affect some of the findings/conclusions.

Additional comments

In general, I'm a big fan of studies that set out to identify proper reference genes for RT-qPCR studies in a particular species under specific growth conditions, so I welcome this contribution for an understudied species. I have raised some technical concerns that can be addressed simply by analyzing a few more genes. I feel this will result in a stronger report. I've also asked for a few more details in the method sections, to try to make this paper more MIQE-compliant.

Reviewer 2 ·

Basic reporting

Figure 1 is never mentioned in the text and its legend is lacking information on what different things in the figure represent.
Figure 2: The font style used is too small.
Figure S2: The font style used is too small.

Experimental design

Some of the results (i.e.the V5/6, V4/5, V9/10 etc) are impossible to understand without knowledge about the method.

Validity of the findings

Difficult to judge. In several cases, the authors draw conclusions about which of the genes that are best suited as reference genes for certain conditions, but in my opinion they do not explain how they draw these conclusions

Additional comments

This manuscript describes cloning and evaluation of potential reference genes to be used for qRT-POCR in two genera of Amorphophallus (Amorphophallus Albus and Amorphophallus konjac) in different stress conditions (heat and water-logging) and in different tissues. The rational for the experiments is that Amorphophallus is a herb which is used to treat different chronic diseases, but genomic data is limiting. Since Amorphophallus is a shade-demanding plant and therefore sensitive to heat stress and waterlogging, it is important to determine expression levels of genes associated with resistance to stress. While this might be a project worth pursuing, I think that there are several errors, mistakes and results that are not described properly. This makes it very difficult to evaluate the quality of the presented data. My concerns are listed below.

Line 38-39: The authors write “….housekeeping genes are less subject to the environment and undergo the same variations as the target genes throughout the course of the experiment.” Really? If they undergo the same variations as the target genes, they would not be good as reference genes. As far as I understand, a good reference gene should not undergo variations in response to the environment.
Line 42: The authors write that ideal reference genes should not have any pseudogenes. That might be a valid argument, but I don´t understand the statement that pseudogenes are bad because they could inhibit amplification of genomic DNA. Why would inhibition of amplification of genomic DNA be bad for qRT-PCR?? You do not want to amplify genomic DNA when you do qRT-PCR so anything that would inhibit that would be good....
Line 97-98: The authors write that they have designed degenerate primers based on conserved regions of homologous sequences of different species. What are these species?
Line 98. The authors refer to Table 1 which describes different candidate reference genes. One of these is labelled “his” in the table. Which his gene do the authors mean? There are several.
Line 101: The authors refer to Table 2. The “title” of column 4 in the table is “Source”. What does that mean? Did the authors perform the PCR on genomic DNA from those plants (Oryza sativa and Zea mays)?
Line 127: The authors write that their specific primers for candidate reference genes are 100-200 bp in length. First, a primer is single stranded and is therefore not composed of base pairs, but rather of nucleotides. Secondly, I doubt that the primers are that long. The authors probably mean the length of the expected amplified DNA sequence.
Line 128: “single peak curve” should be “single peak melting curve”
Figure S1A: The font in the figure is too small.
Line 135: The value 92.8% is wrong. According to table 3, the E-value for the RP gene is 92.4%
Lines 138-144: This section is confusing. The author makes a reference to figure S2 on line 140, but it should probably have been to figure 1. Figure 1 is not mentioned anywhere in the entire manuscript. The reference to figure S2 should probably fit better after the word “(CYP)” on line 138. In addition to never being mentioned in the text. The legend to figure 1 is lacking a lot of information about what the different colored circles and the boxes represent. The authors also mix mean Ct values and median Ct values. On line 140 they write that the Ct value for TUB is 27.96 and on line 141 they write that the threshold fluorescence for TUB is also 27.96. Do they mean that the Ct value and the threshold fluorescence is the same? On line 140 the authors also write that the Ct value of the 18S gene is significantly higher than the others, but as I read Fig. S2 the Ct value for the 18S gene is significantly lower than for the other genes.
Tables 4-6: The authors never explain what the bold style of some genes should indicate.
Fig.2 The font used in the figure is too small.
Line 164: The authors write “In the total set, EIF4A and EF1-a were the two best reference genes.” That might be true, but the authors never describe how they calculate the values in the column of figure 4 labelled “Total”. That is the same also for the corresponding columns in tables 5 and 6.
Line 165-174: It will be very difficult for readers to understand what the different values of V5/6, V4/5, V9/10 etc means if they are not familiar with the method.
Line 181-182: The authors write “In Amorphophallus albus under heat stress, EF1-a was the best reference gene, followed by EIF4A.” Why is UBQ not good? It is ranked above EIF4A.
Line 195-196: The authors write “Under heat stress, EF1-a and H3 were the best reference genes in Amorphophallus albus and Amorphophallus konjac, respectively.” Why is not EIF4A? It is top ranked in both Aa and Ak after heat stress.
Line 202: The heading “Discussion” is miss-spelled as “Disscussion.” In addition, the discussion section is far too long and is mostly a repetition of things already described in the introduction and result sections.

Reviewer 3 ·

Basic reporting

The MS entitled “Cloning and evaluation of reference genes for quantitative real-time PCR analysis in Amorphophallus” by Kai Wang and coworkers intends to determine and evaluate the stability of reference genes for future gene expression studies in Amorphophallus. The absence of genome was overcome by PCR amplification and cloning of 10 reference genes based on reports from other plant crops (although the authors mentioned that there is some transcriptome data available). The stability of reference genes was assessed with the most commonly used software available today.

In general terms, the MS is well written and clear. Nevertheless:

1. Line 10: the authors should refer to quantitative real-time PCR as “RT-qPCR”, since the quantitative nature of the employed methodology is associated to the PCR reaction and not to the reverse transcription reaction. Please, correct throughout the text.
2. Line 14: The reference to “housekeeping genes” is obsolete. The authors should refer to “reference genes”.
3. Lines 12-15. The authors are encouraged to indicate that the set of reference genes are intended for their use in “Amorphophallus”. Please, also provide a short description of the plant, since it is very uncommon plant specie.
4. Line 17: The authors refer to two abiotic conditions: heat and cold. In line 19-20, heat and waterlogged. Please, clarify and correct accordingly. For example, in line 76, the authors mention “both stressors”.
5. Line 28: It is possible to analyze several regulatory networks in biology. Are the authors taking about “transcriptional regulatory networks”? Please, correct accordingly.
6. Line 31: please, change “rapidity” by “great speed”.
7. Line 33: after the citation Chang et al., 2012, please provide appropriate citation to a couple of reviews describing some of the common pitfalls associated to RT-qPCR.
8. Line 36: I disagree with the authors: the term “housekeeping gene” is a rather outdated term (see my comment above). While reference genes should be proven as stable in terms of expression in the conditions under study (as intended in this MS), reference genes should represent highly expressed genes as well as low or moderate expressed genes. The reference Thellin et al., 1999 is old and outdated. Please, modify.
9. Line 38-39: The authors should carefully considered sentence revision. For example: “Reference genes are expected to be less affected in terms of their expression levels upon environmental perturbations, but may undergo similar variations as target genes throughout the course of the experiment (CITATION). Consequently, proper validation of reference genes is required.”
10. Lines 42-43: The sentence related to pseudogenes needs clarification.
11. Line 44: The authors are encouraged to be more precise when describing “associated problems”.
12. Line 46: “in vitro” (in italics).
13. Line 53: Please provide full name for “O. javanica”. It is the first time mentioned in the manuscript.
14. Line 53: replace “housekeeping genes” to “reference genes”. Please, correct this throughout the text.
15. Lines 67-69: The sentence is confusing: while the plant is not resistant to heat or excessive water, the authors highlight the need of validated reference genes associated with resistance. Please, modify the sentence accordingly.
16. The paragraph between lines 58 and 70 should appear in the text much earlier. The absence of genome for Amorphophallus, as well as its importance in Asia, justifies the need of reference genes. Please, modify the text accordingly.
17. Line 80: genera or species?
18. Line 82: please provide information of the substrate used to cultivate plants.
19. Line 89: After the word “treating”, please insert the following: “the obtained RNA”.
20. Line 91: the absence of contaminants (purity) can be determined by employing a spectrophotometer. Consequently, the word “quality” on line 91 is incorrect. RNA quality (or integrity) was assessed by electrophoresis (mentioned in line 93).
21. Line 97: How many reference genes were analyzed? Nine or ten?
22. Line 98: please, provide the name of the species.
23. Line 98-99: The sentence “The PCR products of the degenerate primers were…” needs revision.
24. Line 99: please provide more information of the ligation procedure. For instance, which vector was used? “E. coli” should be in italics.
25. Line 100: a basic Google search of “Basic Graphics Interface (BGI, China)” returned the “Beijing Genomics Institute”. Please, verify and correct accordingly.
26. Line 112: the delta (Ct) method is NOT used to analyze the stability of reference genes. Please, correct. Please, also provide the reference for each software, and not just a single paper.
27. Line 120: after the word “between” please change the sentence to “all pair of genes”.
28. Line 125: Remove the word “The” at the beginning of the line. Provide appropriate reference for the Primer3Plus software.
29. Line 128: after the work “peak” insert “melting”.
30. Line 140: the Ct(s) values of the 18S gene were significantly lowered (and not higher) as stated in the manuscript. Please, correct the sentence.
32. Line 186: replace “all genomes” by “the two plant species analyzed in this work”.
33. Line 189: “Michael” is the first name. Please, correct to the author´s last name.
34. Line 215: to which resistance are the authors referring to?
35. Line 231: replace “genera” by “specie”.

36: Regarding Figure 1: What is the meaning of the blue dot within each box?
37: Regarding Table 1: The title needs to refer to “reference genes” for RT-qPCR. Please, correct. The same is true for Table 2.
38: Table 2: The word “Identities” needs to be replaced by “Identity”.
39: Table 3: Please, make consistent the gene names across tables. Also, please provide the method used to calculate the Tm value (and citation if needed).
40: Table 4: Please, indicate the meaning of the bold type used in the table. The same observation is valid for Table 5.

Experimental design

Two major concerns:

1. geNorm assumes that the reference genes chosen in the analysis are NOT related in terms of function. In order words, that two selected reference genes do not encode for a protein that participates in the same biological function (please, see lanes 239-243). Not surprisingly then, both elongation factors were reported as the most stable reference genes. The authors are encourage to perform at least one of these two options: 1) please, repeat the analysis by removing the genes that are functionally related to each other (caveat: the number of total reference genes will be reduced); 2) replace some of these genes by another putative reference genes, in order to greatly improve the results.

2. Based on the data presented in Figure 3, the number of reference genes that the authors “need” to use for appropriate normalization of RT-qPCR data is rather big. For example, as mentioned in line 166, 5 reference genes are needed (as stated by the authors). When looking a Figure S2, you can easily observe that the Cts of the 18S gene range from 30 (low expressed) and below Ct=5 (very high quantities). I am wondering if such a big variation is compromising the validity of the results. The significant variability strongly suggests that other genes are required to extend the validity of the results. Moreover, there might be technical issues in the results reported: all qPCR machines need to determine two critical values (baseline and the cycle threshold). When Cts are as low as 5, basically there is no “space” in the amplification curve to determine the cycles that are going to be used as baseline (usually between cycles 5 and 13).

Validity of the findings

I mentioned (see above) two major concerns. For instance, based on the very low Cts of the gene 18S, any conclusion is simply not valid.

To make data more robust, simply remove some genes (e.g. 18S), repeat the analysis and include at least two additional reference genes.

Additional comments

Most researchers simply do not take any consideration on correct RT-qPCR normalization. The use of not validated reference genes or simply a single gene is widespread in the literature. So, the considerations taken by these authors speaks highly of their attitude towards well-done science.

---

## Round 0.2 · Major Revisions

· Academic Editor

Major Revisions

I sent this back to two of the original reviewers and took a look myself.

As you will see the two reviewers have again raised substantial suggestions for improvement which are detailed below. We will consider publishing your manuscript only if you can accommodate their suggestions in a revised version or explain satisfactorily why their comments are invalid.

Beyond the technical issues raised, we strongly suggest that you have a native English speaking colleague review your manuscript to improve the ongoing language issues.

Reviewer 1 ·

Basic reporting

1) The English needs thorough revision. As a non-native speaker myself, I know how challenging this can be.

There are several spelling and grammar issues, to the point that they distract from the science. I can’t possible list all of them, but I would urge the authors to show their manuscript to a native English speaker or use a manuscript editing service. In its current form, it is difficult to envision the MS being considered further.

2) Many of the references related to RT-qPCR are not accurate and do not highlight the most representative/authoritative papers in the field, but a very biased selection based on crops.

Line 49.
Replace ALL of those references, for instance with :

J. Mol. Endocrinol. 25, 169–193 (2000).
Clin Chem. 2009 Apr;55(4):611-22.
Nat Protoc. 2006;1(3):1559-82.

Line 50:
Replace Chen et al., 2011 with Nucleic Acids Res. 2011 May;39(9):e63.
Line 52:
Replace Chan et al 2012 with Methods Mol Biol. 2014;1160:19-26.

Line 67:
Replace Chandna et al., 2012, with Methods Mol Biol. 2014;1160:19-26.
Line 281:
Replace Kong et al., 2013 with J. Mol. Endocrinol. 25, 169–193 (2000).

Line 282:
Replace Huang et al., 2014 and Zmienko et al., 2015 with Methods Mol Biol. 2014;1160:19-26.

3) The authors alternate between saying they use either 8 (Line 124, 132) or 9 (Line 95, 170) candidate genes. Which one is it?

4) Please make sure all species names are written in the proper format. In many cases it says A. Albus instead of A. albus.

5) Unless you actually show the difference in Cq values between doing a PCR on the RNA or on the cDNA, you can’t say that “DNA contamination was removed”. Line 116 should read something like “RNA was treated with RNAse-fee DNAse I according to manufacturer instructions”. Nothing else.

6) Please state that you used random hexamers as a primer strategy.

7) Where were the programs downloaded from? (Line 148)

8) The variations in Cq levels between samples do not necessarily mean that a gene is not stable; until the samples are normalized, that variation can simply reflect technical (and not) biological variation.

Hence, line 186 should either be changed or removed.

9) Line 197. Being below the M value of 1.5 does not mean that a gene can be used on its own as a reference gene. The next step in geNorm, determining how many genes are required, is crucial.

10) Line 203. I can’t understand what the authors mean by “8 and 5 candidate reference genes could be used as the reference genes based on the principle of geNorm in A. albus and A. konjac, respectively”

11) Line 207. UBQ was not the most unstable in A. konjac; it was GAPDH.

12) Line 208. CYP has a value greater than 1.5

13) Line 214. “At the same time, the lowest value represents the best combination when no value meet this prerequisite of cut-off”. That is not correct. If your samples do not reach the cutoff, that does not mean that the closest one is OK. It only means that under these conditions, no combination of these genes can be used as a proper set of reference genes.

14) Line 218. The implication behind this is incorrect. While it is of course better to use more reference genes that a single one (Vandesompele et al., 2002), it is simply not correct to imply that we should just use 2 or 3, without any analysis, and expect the data to be accurate. If your analysis suggests that 5-7 of the genes *you tested* are required for normalization, then that’s it. Such a high number simply suggests that the genes chosen are not very good for these conditions in the first place, and that other (additional!) genes should be tested, until the number of genes required goes down to a practical number. This is regardless of the crop references the authors provide on the rebuttal letter.

15) Line 224. There’s a typo after “Normfinder”.

16) What’s the cutoff used for the NormFinder algorithm?

17) Line 327. There should be data supporting this recommendation. One cannot simply combine genes and expect that they work properly. One must define the identity and number of proper reference genes for certain condition.

18) Line 122. The authors mention that they used “*up* to 1 ug of total RNA”, which implies that they didn’t they use the same amount of RNA for all samples. Is this the case? This is a very important element to keep in mind when trying to minimize technical variation in gene expression measurements via RT-qPCR

19) Please include in the MS the dilution performed on the cDNA prior to PCR.

20) Please make sure all instances of qRT-PCR are changed to RT-qPCR.

Experimental design

Major concern

My main criticism still stands: the candidate genes the authors chose includes genes that participate in a related biological process, something that should be avoided, as it can artifactually suggest that those two related genes are the most stably expressed under certain conditions. The authors used two genes involved in translation and those same two, are suggested to be the most stable. In my previous review, I recommended that they excluded one of those, added a new gene, that plays a role in an unrelated process, and re-do the analysis. While it is formally possible that those two are indeed the most stably expressed ones, the authors cannot discard the possibility that they appear to be the most stable due to their participation in the same process. At this point, I suggest to either replace one of those translation-related genes with another gene not involved in translation or any other process already represented among the candidate genes and re-do the analysis, or include a phrase in the discussion saying that while the analyses suggest that the two translation-related genes appear to be the most stable, the authors can’t formally exclude that they appear to be so due to them being co-regulated due to participating on a related biological process. They could even make a case by mentioning the results of the other tools.

The authors make some attempt to justify the use of these genes on Line 310, but their argument lacks support.

The response to this criticism in the rebuttal letter and in the manuscript, did not address the concern.

Minor:

1) Why did the authors use the 2-{delta}{delta}Cq method, which assumes similar efficiency values between target and reference genes, when they have already determined the precise efficiency values? They should use those experimentally determined ones for the calculations. See https://www.ncbi.nlm.nih.gov/pubmed/17291332.

Validity of the findings

I've expressed my concerns in the previous section, which may affect some of the findings/conclusions.

Inclusion of the analysis of SHSP expression is a nice addition, but heat shock is one of the few conditions in which the selected genes operate well and in a reasonable number. There are other conditions in which the numbers are not practical and again, saying “just use 2-3” is not a proper argument. I think that the authors should simply acknowledge and state that for those conditions (e.g. waterlog in Ak), the selected candidate genes simply vary too much to be useful in a practical manner and refrain from making any suggestion/recommendation of using 2-3 of those genes. The selected candidate genes are simply not good for those conditions.

Reviewer 3 ·

Basic reporting

The manuscript represents an improved version of the previously submitted manuscript. It is relatively unambiguous, but still needs substantial revision from a gramatical point of view. There are several sentences that can just be better written. The authors are encourage to proof read the MS by a english-native speaker or any professional writer.

As an enhanced version (second version of the MS), now the authors highlight the importance of the plants been studied. So, in general terms, the MS now reads more easily, starting from highlighting the plants key characteristics and then, moving towards the specific performed analysis. Literature was improved, although some minor errors were still identified.

The structure of the MS meet general standards. Still, in the results sections, the authors simply describe which genes are better reference genes, for each plant/condition, and for three different softwares. From that point of view, the reading is realtively harsh and repetitive.

Experimental design

As I mentioned in my previous report, the MS intends to address a very fundamental issue relative to RT-qPCR that most authors simply do not pay attention when publishing gene expression data. I would like to highlight this observation from this group of authors.

In this opportunity, the authors removed data that at first glance, looked to me not meaningful (18S data). Since most readers are not familiar with many of the terms employed by the three algorithms used in this MS (e.g. GeNorm), in this opportunity, the authors described several of them in a more understandable fashion.

Many of the selected reference genes are, from a functional point of view, related to each other. Not surprisingly then, the best two reference genes are related to each other, from a functional stand point. Nevertheless, the authors conclude that "we suggest use 2 or more stable reference genes involved 
in different biological processes for normalizing the gene expression.". Consequently, it is very hard to understand the experimental design and the conclusion.

Validity of the findings

At least for a heat treatment, now the author analyze the expression of a "marker gene" related to that particular stress, when normalized using EF1a, EIF4A and other reference genes. Not surprisingly, no big differences were observed for the two mentioned reference genes (independently, or in combination). On the other hand, when two normally employed reference genes are used (actin and GAPDH), significant differences were observed upon stress. So, the data provided by the authors exemplifies that the result of a particular gene in terms of its mRNA levels can be greatlly impacted depending on the chosen reference gene(s). Although the two "best genes" are functionally related, a third "good gene" provided a similar result in comparison with the two "best genes". Maybe, the authors can highlight this observation in the final version of the manuscript.

Additional comments

Minor comments to the authors:

1. Replace “which usually used as reference genes” to “which are usually used as reference genes”.
2. At the end of the abstract: replace “…the two best reference genes could be used to…” to “…the two best reference genes that could be used to…”
3. In general terms, the grammar within the abstract needs extensive revision.
4. Line 44: “Introduction”, not Introducqion”
5. Line 51: replace “…using a set of standardized experimental conditions are also two important factors for…” to ““…using a set of standardized experimental conditions are among the most important factors for…”
6. Line 52: RT-qPCR, and not qRT-PCR. Importantly, while the authors indicate that this and related issues were modified, this seems not to be the case. The same is observed in the legend of Figure 1.
7. Line 62-63: remove the following sentence: “Moreover, in order to avoid amplification of pseudogenes there should be no genomic DNA present (Yeap et al., 2014). (Remove citation if requiered).

8. Line 91: between words “genes” and “contribute”, insert the word “may”.
9. Line 91-92: Change the sentence “However as one of main identification methods of gene expression,…” to the following: 
“However as one of main methods of gene expression determination,…”
10. Line 93. Remove the word “the” at the beginning of the line.
11. Line 99: after the word “which” insert “is”.
12. Line 105: define KGM.
13. Line 105: “albus” not, “Albus” as written in the MS.
14. Line 124+126: the authors should include the exact list of plant species used for degenerate primer design.
15. Line 127: clarify the word “connection”. Do you mean, “cloned products”?
16. Line 130: after liquid, insert the word “cultures”.
17. Line 157-158: the text between these two lines allows me to exemplify that the MS needs English revision: it is written “Small heat shock proteins (SHSP) were the important proteins in plants and had the function of molecular chaperone”, while it should be “Small heat shock proteins (SHSP) are important proteins in plants that function as molecular chaperones”
18. Line 162: please clarify the meaning of “two kind combinations”.
19. Line 175: Zhu et al., 2013 is NOT the original and first citation of the method. Please, correct.
20. Line 203:
21. Line 207: Verify punctuation.
22. Line 224: remove the word “speciestes” and correct with the appropriate word.

23. Line 295: Do you mean four reference genes, no?

---

## Round 0.3 · Minor Revisions

· Academic Editor

Minor Revisions

Thank you for submitting your manuscript for consideration by PeerJ. It has now been seen by one of the original referees again whose comments are enclosed.

As you will see the referee supports publication of your work in PeerJ; however, before I can go on to officially accept your study and transfer the manuscript files to our production team I have to ask you to address the points raised by the referee in a final revision.

I am therefore formally returning the manuscript to you for a final round of minor revision. Once we should have received the revised version, we should then be able to swiftly proceed with formal
acceptance and production of the manuscript.

Reviewer 1 ·

Basic reporting

I was pleased to read this revised version of the MS. It has been significantly improved and I think it makes the point clearer that in previous versions.

I still think that the English should be revised, but it I acknowledge that it was improved from the previous version of the MS.

Some suggestions:

Line 46. Change Xu et al, 2011 for something more appropriate, maybe PMID: 26355593.

Line 50. I think a more appropriate reference for talking about efficiency than Huang et al., 2014, is Pfaffl, M.W. Chapter 3: Quantification strategies in real-time PCR, A_Z of quantitative PCR, Bustin S.A. (Ed) International University Line, La Jolla, CA (2005)

Line 56- 57. I would change that statement to “These genes are ubiquitously expressed and are typically involved in housekeeping processes.” The second part “simultaneously, reference…”, should be removed, as that claim is made in the following paragraph.

Line 62: For clarity, I would replace “are associated with defects” with “do not typically satisfy all these criteria”

Line 75. My understanding is that the genus is Amorphophallus. I don’t understand why the authors say the genus is “A. Blume”

Line 85. I would change the phrasing of that statement slightly: “Interestingly, Amorphophallus is a typical shade-demanding plant, intolerant to heat damage and waterlogging, and flooding and high temperature are two important factors that significantly impact Amorphophallus growth".

Line 96. Please mention that SHSP is upregulated in response to heat stress in plants, instead of saying that “is related to resistant to heat stress in plants”

Line 97. Replace “of” for “to”

Line 115. What’s the DNaseI brand?

Line 122. Replace qRCR by qPCR

Line 126. There’s an “et al” in that line that I think shouldn’t be there

Line 147. Reference should be corrected, from De et al., 2105 to De Spiegelaere et al., 2015.

Line 148. Software versions should be included

Lines 160-162 should be in the results. Maybe in line 260?

Line 175-176. I think saying “using a standard curve generated by 10-fold serial dilutions of gel-extracted PCR products” is clearer.

Line 203. Change “expect for” for “besides”

Line 203. It appears that all 9 genes could be used for waterlogging conditions in A. albus

Line 208. Add quotation marks and capital letter to total (so that it reads “Total”).

Line 221. I think this line would be clearer with a little change: “While the 0.15 threshold was not met in the other samples, the geNorm developers emphasized in the user manual, that the proposed threshold of 0.15 must not be taken as a strict cutoff. The cut-off value was set only to offer guidance for determining the optimal number of reference genes”.

Line 230. I would change to “has the drawback that it identifies only a single best reference gene”.

Line 247. Can the authors find any other examples in the literature, showing that the results from geNorm and Normfinder are usually consistent with each other, but differ from those given by Bestkeeper?

Line 249. In Aa, ACTB ranked second under heat stress. In Ak, it was EF1-alpha.

Line 268. Remove “novel”


Minor comments:

Is there a list/table with the 78 samples used?

Legend of Fig2 says “groupsas”

Please remove the bold from the tables; as the genes are ranked, putting some genes in bold does not add additional information.

Experimental design

Major:

1) In the discussion, the authors discuss at length the issue of using genes belonging to the same biological process as reference genes (Lines 330-345), which is an issue raised by various reviewers. While this discussion is greatly appreciated, there is an issue. What I recommended to do when I read a previous version of the MS, was to leave out (exclude) one of the translation genes, say EIF4A, as if they had never tested it, and *repeat the geNorm analysis*. If again they see EF1-alpha as one of the most stable ones, then it probably behaves as a stably expressed gene and it is not because it is being co-expressed with another gene due to being part of the same biological process. This will be the strongest evidence against the argument that the two translation-related genes are being suggested to be the most stable by geNorm, due to being in the same biological process. I think this should be done, particularly because the authors insist on recommending those two genes.

2) The authors mention in the discussion (and in the conclusion) that their results show “that using multiple reference genes can improve accuracy of the results”, and that “a single reference gene could not normalize the expression well”, but their results don’t show that. In fact, their results show that you can get almost identical results by using a single reference gene. This is seen in Fig 4. (compare results using UBQ and EF1-alpha+UBQ). To get an idea of what I'm saying, please see Fig 4. of this BioRad technical bulletin (http://www.gene-quantification.com/miqe-case-study-bio-rad-6245.pdf). In that example, you can see that there is statistical evidence that using more than one reference gene can improve the results.

3) The authors used SHSP as a gene that should go up under heat stress, but it wasn't known if that happened in Amorphophallus as well as in other plants. As there's no "gold standard" or orthogonal approach to which to compare, the results in Fig 4 do not necessarily show that their identified reference genes allow to see the actual gene expression changes, but only that using poor genes result in different results from those using stable genes.

Minor:

Line 264. Were the differences in gene expression calculated by the appropriate references genes and the ones determined using ACT or GAPDH, statistically significant?

Validity of the findings

No comment.

Additional comments

I think the authors did a great job incorporating the suggestions by the reviewers and as a result, the manuscript is stronger and clearer. By doing some small changes in the MS and a simple re-analysis, the paper will be more sound and will have addressed all concerns raised during the review process. I think the manuscript has been greatly improved.

---

## Round 0.4 · accepted · Accept

· Academic Editor

Accept

I am happy with the final modifications to the manuscript.